# Chronic Systemic Dexamethasone Regulates the Mineralocorticoid/Glucocorticoid Pathways Balance in Rat Ocular Tissues

**DOI:** 10.3390/ijms23031278

**Published:** 2022-01-24

**Authors:** Marta Zola, Dan Mejlachowicz, Raquel Gregorio, Marie-Christine Naud, Frédéric Jaisser, Min Zhao, Francine Behar-Cohen

**Affiliations:** 1Centre de Recherche des Cordeliers, Sorbonne Université, Université de Paris, Inserm, From Physiopathology of Retinal Diseases to Clinical Advances, 75006 Paris, France; marta.zola@aphp.fr (M.Z.); mejlachowicz.dan@hotmail.fr (D.M.); raquelgregorio.ufmg@gmail.com (R.G.); marie-christine.naud@crc.jussieu.fr (M.-C.N.); frederic.jaisser@gmail.com (F.J.); elodiecn@gmail.com (M.Z.); 2Assistance Publique-Hôpitaux de Paris, Department of Ophthalmology, Ophtalmopôle, Hôpital Cochin, 75014 Paris, France

**Keywords:** retina, central serous chorioretinopathy, glucocorticoid, mineralocorticoid

## Abstract

Central serous chorioretinopathy (CSCR) is a retinal disease affecting the retinal pigment epithelium (RPE) and the choroid. This is a recognized side-effect of glucocorticoids (GCs), administered through nasal, articular, oral and dermal routes. However, CSCR does not occur after intraocular GCs administration, suggesting that a hypothalamic-pituitary-adrenal axis (HPA) brake could play a role in the mechanistic link between CSCR and GS. The aim of this study was to explore this hypothesis. To induce HPA brake, Lewis rats received a systemic injection of dexamethasone daily for five days. Control rats received saline injections. Baseline levels of corticosterone were measured by Elisa at baseline and at 5 days in the serum and the ocular media and dexamethasone levels were measured at 5 days in the serum and ocular media. The expression of genes encoding glucocorticoid receptor (GR), mineralocorticoid receptors (MR), and the 11 beta hydroxysteroid dehydrogenase (HSD) enzymes 1 and 2 were quantified in the neural retina and in RPE/ choroid. The expression of MR target genes was quantified in the retina (*Scnn1A* (encoding *ENac-**α*, *Kir4.1* and *Aqp4*) and in the RPE/choroid (*Shroom 2*, *Ngal*, *Mmp9* and *Omg*, *Ptx3*, *Plaur* and *Fosl-1*). Only 10% of the corticosterone serum concentration was measured in the ocular media. Corticosterone levels in the serum and in the ocular media dropped after 5 days of dexamethasone systemic treatment, reflecting HPA axis brake. Whilst both GR and MR were downregulated in the retina without MR/GR imbalance, in the RPE/choroid, both MR/GR and *11β-hsd2*/*11β-hsd1* ratio increased, indicating MR pathway activation. MR-target genes were upregulated in the RPE/ choroid but not in the retina. The psychological stress induced by the repeated injection of saline also induced HPA axis brake with a trend towards MR pathway activation in RPE/ choroid. HPA axis brake causes an imbalance of corticoid receptors expression in the RPE/choroid towards overactivation of MR pathway, which could favor the occurrence of CSCR.

## 1. Introduction

Glucocorticoids (GC) remain amongst the most widely prescribed class of drugs in many fields of medicine for their potent anti-inflammatory effects. The early recognition of the severe side effects induced by cortisol [1] has led to the development of synthetic or semi-synthetic cortisol derivates with increased anti-inflammatory effects and reduced mineralocorticoid side-effects [2]. Indeed, hypertension and hydro-sodium retention have been associated with the activation of the mineralocorticoid pathway [3] while beneficial anti-inflammatory effects have been associated with the activation of the glucocorticoid receptor, leading to the classical potency glucocorticoids classification table [3,4,5]. 

The ocular side effects, such as cataract [6] and glaucoma [7,8] have also been recognized at early stages [9] and are still associated with both systemic and local routes and are not alleviated by synthetic GCs. Another less frequent ocular side-effect of GCs is central serous chorioretinopathy (CSCR), a chorioretinal disease characterized by the occurrence of serous retinal detachments (SRD) secondary to a focal disruption of the retinal pigment epithelium (RPE) barrier and often associated with detachments of the pigment epithelium (PED) and an increased choroidal thickness [10,11]. However, in contrast to the other GCs-induced side effects, the route of GCs administration influences the risk of CSCR. The nasal, articular, oral and dermal routes favor CSCR [12,13,14,15,16,17,18,19,20,21] but not the high and sustained intraocular GCs [22,23]. This clinical observation led us to hypothesize that it is not the increase in ocular GCs levels but rather a consequence of systemic corticosteroids on ocular corticosteroids that may be the link between CSCR and corticosteroid therapy.

Cortisol binds with a similar affinity to both glucocorticoid receptor (GR) and to MR [2] and in the eye, as no cortisol binding globulin is present [24], the availability of free cortisol mostly relies on the cortisol/cortisone balance that is regulated by the activity of the 11-beta hydroxysteroid dehydrogenase type 1 and 2 enzymes (HSD-1 and HSD-2) [25,26]. GCs also activate MR at various intensities depending on the molecules, the drug doses and on the level of expression and activity of MR in specific cells and tissues. In non-classical MR sensitive tissues, MR activation can exert deleterious effects, as shown in the brain [27], the vessels, the skin [28], the heart [29,30] and the retina [11,31,32]. Mineralocorticoid receptor (MR) pathway activation by GCs or by cortisol has been proposed as a potential mechanism linking GCs with CSCR since aldosterone and/or MR pathway activation in animal models induce features of CSCR, although they do not completely recapitulate the human disease [11,32,33]. However, there is no clinical evidence that MR is overactivated in eyes with CSCR.

The reason why intraocular GCs is not a risk factor for CSCR as compared to systemic administration remains unexplained and whether the hypothalamic-pituitary-adrenal axis (HPA) brake and subsequent adrenal insufficiency, that is not induced by intraocular GCs could be involved in CSCR pathogenesis has not been questioned. In the present study, we studied the corticosterone ocular levels and the corticoid receptors balance upon a brake of the hypothalamic-pituitary-adrenal axis after systemic dexamethasone treatment in rats. Our results indicate that systemic GCs and endogenous corticosterone differentially influence the expression of the gluco- and mineralocorticoid receptors in different ocular tissues and promote an imbalance towards a hyper activation of the mineralocorticoid pathway in RPE/choroid. Potential link with CSCR is discussed. 

## 2. Results

### 2.1. Correlation between Serum and Ocular Corticosterone and Dexamethasone Levels

The mean corticosterone serum level in control untreated rats at baseline was 21.2 ± 6.5 ng/mL [14.7–32.04, *n* = 8]. As expected [34], after 5 days of dexamethasone systemic injection, the serum corticosterone level significantly dropped to 1.2 ± 1.4 ng/mL [0–3.8, *n* = 16, *p* < 0.0001]. Interestingly, in saline injected rats, serum corticosterone also dropped to 13 ± 2 ng/mL [10.1–14.5, *n* = 4], a level that was not significantly lower than baseline, but also not significantly different from the corticosterone level at day 5 in rats treated with dexamethasone (*p* = 0.007) (Kruskal–Wallis test *p* < 0.0001, followed by Dunn’s comparisons) (Figure 1A).

In ocular media, mean corticosterone level at baseline was 2.9 ± 1.2 ng/mL [1.3–5.4, *n* = 7]. Significantly, it dropped to 0.4± 0.4 ng/mL [0–1.3, *n* = 8], 5 days after systemic dexamethasone treatment, paralleling the serum drop (*p* < 0.002). In saline treated rats, mean corticosterone ocular level non significantly dropped to 1.45 ± 1.3 ng/mL [0–2.9, *n* = 4] (*p* = 0.46). Ocular corticosterone levels were not significantly different in saline and dexamethasone treated eyes (*p* = 0.5) (Kruskal–Wallis test *p* < 0.0006, followed by Dunn’s comparisons) (Figure 1B).

There was a significant correlation between corticosterone serum levels and ocular levels (Spearman correlation, r = 0,76, *p* = 0.0001), the ocular corticosterone level being around 10% of the serum level (Figure 1C). The mean dexamethasone serum level after 5 days of dexamethasone systemic administration was 60.1 ± 18.9 ng/mL [18–83.9, *n* = 14] and the ocular mean level was 10.7 ± 4.09 ng/mL [6.6–17.6, *n* = 7] (Figure 1D). There was no significant correlation between dexamethasone ocular and serum levels, although a tendency towards an inverse correlation was observed (Spearman correlation, r = −0.28, *p* = 0.55) (Figure 1E).

### 2.2. Corticosteroid Receptors and HSDs Expression in Ocular Tissues after Dexamethasone Treatment

In the neural retina, at 5 days after systemic treatment, *Nr3c1* encoding GR expression was significantly reduced both after the administration of saline (*p* < 0.001) and dexamethasone (*p* < 0.01) as compared to untreated rats. Expression of *Nr3c2* encoding MR expression was also significantly downregulated in saline and dexamethasone groups (*p* < 0.001). The *Nr3c2*/*Nr3c1* (referred as MR/GR) expression balance did not change significantly. In the retina of rats treated with dexamethasone, a significant down-regulation in *11β-hsd1* (*p* > 0.01) and an up-regulation in *11β-hsd2* gene expression (*p* < 0.001) was measured. A non-significant trend towards a similar regulation was observed after saline treatment (Figure 2A).

In the RPE/choroid complex, the non-significant up-regulation of *Nr3c2* expression translated into a significant increase in the *Nr3c2/Nr3c1* ratio in eyes of rats treated with dexamethasone (*p* < 0.01). Whilst *11β-hsd1* expression was down-regulated in the RPE/choroid from saline treated rats (*p* < 0.01), *11β-hsd2* expression was significantly up-regulated in dexamethasone-treated rats (*p* < 0.0001) (Figure 2B) (analyses were performed using the non-parametric Kruskal–Wallis ANOVA test followed by a Dunn’s comparison test). 

Taken together, these data indicate that in the RPE/choroid from rats treated with systemic dexamethasone, the transcriptional regulations of corticoid receptors and HSD enzymes favor a balance towards an activation of MR pathway and a reduction of GR pathway activation. In rats treated with saline, a similar trend is observed suggesting that these regulations could result, at least in part, from reduction of serum corticosterone levels in both saline and dexamethasone treated rats. In saline-treated rats, the very low corticosterone ocular levels together with a down-regulation of GR indicate a low activity of glucocorticoid pathway. 

In the iris/ciliary body complex, only dexamethasone treatment induced a significant down-regulation of *Nr3c1* (*p* < 0.01) and a higher *Nr3c2/Nr3c1* ratio, together with a significant increase in *11β-hsd2* expression (*p* < 0.01) (Figure 3) Saline treatment did not induce significant changes in *Nr3c1*, *Nr3c2*, *11β-hsd1* or *11β-hsd2* expression. 

### 2.3. Regulation of MR-Induced Genes in the Retina and RPE/Choroid

To evaluate whether the imbalance of expression in the *Nr3c2/Nr3c1* and *11β-hsd1/11β-hsd2* ratio translate into transcriptional consequences, we have quantified the expression of genes known to be regulated by GR/MR pathways in the neural retina such as *Scnn1A* encoding the Na+ channel ENac-aplha the K+ channel *Kir4.1*, as well as the water channel *Aqp4* [35,36]. As shown in Figure 4, the three genes were significantly down-regulated in the neural retina in rats after 5 days of dexamethasone treatment and *Scnn1A* and *Aqp4* were significantly down-regulated also after saline injection, which is in line with the downregulation of genes encoding both GR and MR after both treatments, but without MR/GR imbalance.

In the RPE/choroid complex, the expression of *Plaur* (*p* < 0.01) and *Fosl-1* was significantly up-regulated after dexamethasone treatment, whilst *Ptx3* was instead down-regulated after saline treatment *Omg* was significantly down-regulated in dexamethasone treated rat (Figure 5). Other genes such *Shroom2*, *Ngal,* and *Ptx3* were up-regulated in dexamethasone-treated rats as compared to saline only (Figure 5) (analysis were performed using the non-parametric Kruskal–Wallis ANOVA test followed by a Dunn’s comparison test).

These results indicate that genes were shown to be regulated by MR pathway activation in RPE cells [37], following the same regulations in the RPE/choroid of dexamethasone treated rats. 

## 3. Discussion

In this paper, we have measured the corticosterone levels in ocular media of rats at baseline and after 5 days of systemic treatment with dexamethasone at 1 mg/kg to mimic a frequent clinical situation, in which systemic corticotherapy causes adrenal insufficiency. At baseline, corticosterone level in ocular media is around 1/10 of the serum level and it correlates well with the serum level. Similar results were found in dogs, in which 61% of the variation in aqueous humor cortisol levels could result from plasmatic variations [31]. Since only the free fraction of cortisol can cross the blood-ocular barriers, it is expected to measure lower cortisol or corticosterone in ocular media as compared to blood. The level of corticosterone measured in the ocular media of rats was in the range of previous studies [24,32]. In the eye, the activity of cortisol is regulated by the HSD-1 and 2 enzymes, that are both expressed in the rat retina, RPE/choroid [33,38] and iris/ciliary body [32]. In the rat, after 5 days of dexamethasone treatment, the hypothalamo-pituitary-adrenal axis brake is identified by the very low level of corticosterone in the serum. Similarly, in the ocular media, corticosterone levels are very low, even below detectable levels in some rats. Corticosterone levels also decreased, although to a lesser extent after repeated daily injection of saline in awaken rats, submitted to repeated restraint stress at a fixed hour for 5 days, which corresponds to a chronic but familiar stressor. Indeed, under such stress conditions, corticosterone was shown to decrease, as compared to an unfamiliar stressor that could have an inverse effect on corticosterone levels [39,40]. Such low corticosterone levels also suggested by low salivary amylase activity in rats submitted to similar stressor [41] indicate changes in the activity of the HPA axis in response of a chronic stress, which could mimic the administration of exogenous GC. Paralleling the serum drop, corticosterone level also decreased in the ocular media. 

In the iris/ciliary body of dexamethasone treated rats, the transcriptional regulations tend to favor a reduction of the GR pathway activation with a reduction in *Nr3c1* expression and an increased *11β-hsd.* In these tissues, more exposed to the systemic circulation, *Nr3c2* was not significantly regulated in dexamethasone or in saline injected rats suggesting that specific regulation of the corticoid receptors expression might occur in various ocular tissues and even cells. 

In the neural retina, the expression of both *Nr3c1* and *Nr3c2* is down-regulated not only in the group of rats treated with dexamethasone, but also in the group of rats treated with saline, indicating that the downregulation does not result from ocular dexamethasone but could result from the decrease in ocular corticosterone levels. In cell cultures as well as in vivo, cortisol and synthetic glucocorticoids were shown to induce the regulation of GR in a differential manner and with different kinetics [42,43,44]. It cannot be discounted that the downregulation of *Nr3c1* in saline-treated rats is a consequence of a transient increase in corticosterone that could have preceded the reduction of serum corticosterone observed at the end of the experiment since a return to the basal state can take several days. Similarly, the mechanisms of MR down-regulation are unclear. In amphibian kidney cells, dexamethasone was shown to down-regulate MR [35] and in LPS-induced uveitis model, we showed that MR down-regulation in iris/ciliary body coincided with high corticosterone levels together with an increase expression of *11β-hsd1* [32] suggesting that both endogenous and exogenous corticoids could also regulate MR expression. On the other hand, in the Goto-Kakizaki diabetic rat model, we observed both a reduced cortisone/cortisol ratio and an increased MR expression in the neural retina [36] showing that the ligand levels and the GR and MR expression levels might not be regulated in a simple manner. In our experiments, *Nr3c2* was also downregulated in the neural retina of rats treated with saline, in which corticosterone levels are reduced in the ocular media and in the serum but in which no dexamethasone could influence *Nr3c2* or *Nr3c1* expression. Other mechanisms such as chronic stress and potential associated inflammation could explain these regulations [37]. 

In the neural retina, the *Nr3c2/Nr3c1* balance remained stable after dexamethasone treatment, although both GR and MR were down-regulated, which was reflected in the down-regulation of genes such as *Scnn1A* (encoding ENac-alpha), *Kir4.1* and *Aqp4* that are up-regulated by both GR and MR activation in glial Müller cells and in rat retinal explants [35]. In dexamethasone-treated rats, the low levels of ocular dexamethasone could have partially counterbalanced the downregulation of *Scnn1A*, *Kir4.1* and *Aqp4.* It cannot be excluded that other mechanisms, unrelated to GR/MR pathway and directly linked to systemic sodium levels, could have played a role in the ion and water channels regulations observed in the neural retina of saline injected rats. This hypothesis requires further explorations. 

In the RPE/choroid complex, the slight *Nr3c2* expression increase in dexamethasone treated rats translates into a significant imbalance towards a higher *Nr3c2/Nr3c1* ratio. The reduced *11β-hsd1/11β-hsd2* balance and the increased *Nr3c2/Nr3c1* indicate a potential MR pathway overactivation in the RPE/choroid complex. To further confirm this imbalance towards MR activation, we tested the expression of genes that we previously identified as MR-target genes in human RPE cells derived from iPSc using an unbiased transcriptomic approach [37]. In the RPE/choroid, *Plaur* and *Fosl1* were upregulated and *Omg* was down-regulated as previously shown by RPE cells, stimulated by aldosterone or by cortisol + GR antagonists to mimic the effects of cortisol on MR [37]. FOS such as 1 AP-1 transcription factor subunit protein controls the expression of *Plaur* (uPAR) [45]. The uPAR is expressed at the basolateral membrane of RPE cells [46] and activation of uPA-uPAR pathways favors choroidal neovascularization [47] and induces RPE epithelial to mesenchymal transition and proliferation [39]. The upregulation of this pathway could be involved in the RPE barrier breakdown, that is, one of the cardinal features of CSCR. Shroom2 is a protein involved in the pigmentation and of RPE cells [48] and pigmentary changes are part of CSCR phenotype. The down regulation of *Omg* in RPE cells, which encodes oligodendrocyte-myelin glycoprotein, could indicate a role for the RPE in the maintenance of choroidal nerves, which control the choroidal blood flow [49] through a secreted isoform of OMG [50]. Alteration of choroidal vasculature is the other cardinal feature for CSCR.

Other genes encoding proteins involved in inflammation and oxidative stress such as pentraxin 3 and lipocalin 2 were mostly down regulated in RPE/choroid from saline-injected rats. Interestingly, we found that systemically low lipocalin levels could be a biomarker for CSCR in cohorts of patients that were not under GCs treatments [51]. Since dexamethasone is a known inducer of pentraxin 3 and lipocalin 2 expression [52], a partial counter-balance could have compensated their reduced expression due to low GR/MR expression in eyes from dexamethasone treated rats.

These results indicate that chronic systemic dexamethasone treatment causes an overactivation of the MR pathway, specifically in the RPE/choroid complex and not in the retina. In a mouse transgenic model that overexpresses the human MR, we observed pathologic changes in the RPE/choroid that mimic the pachychoroid phenotype, without pathologic changes in the neural retina. These results indicate that chronic systemic dexamethasone treatment causes an overactivation of the MR pathway, specifically in the RPE/choroid complex and not in the retina [33,53,54].

We acknowledge weaknesses in our study. Aldosterone could not be measured due to the low volume of ocular media and methods used in this study. Further studies will use mass spectrometry to provide a broader analysis of all neurosteroids and aldosterone in the ocular media. We have not yet validated the regulated target genes at the protein level using either immunohistochemistry or western-blotting, which will be the subject of another study, that will include many other controls including various concentrations of salt in the vehicle.

Taken together, our results show that systemic chronic treatment with dexamethasone, at a dose that induces HPA axis brake, induces variable imbalance in *Nr3c2/Nr3c1* and *11β-hsd1/11β-hsd2* gene expression in ocular tissues, with a trend towards MR pathway overactivation in RPE/choroids, where pathologic changes are observed in patients under glucocorticoids, who develop CSCR. This is in line with the surprising finding that the local instillation of glucocorticoid drops, together with MR antagonists could have beneficial effects in some of severe CSCR patients [40] and could support the beneficial effects of MR antagonists observed by some authors in CSCR patients [55,56,57].

In the ophthalmic literature, stressful psychological events that favor the occurrence of CSCR episodes [11], have been intuitively linked to high systemic cortisol, creating the link with GCs-induced CSCR. Interestingly, none of the studies could identify high serum cortisol levels in CSCR patients and hair-cortisol was not increased in this population [58]. We postulate that under chronic stress conditions, a rumination and depression state, frequently observed in CSCR patients [41], and associated with a reduced activity of the HPA axis [33,59,60], low endogenous cortisol could favor the occurrence of the disease in predisposed patients. In other words, the link between exogenous systemic GCs treatment and psychological stress in patients with CRSC could be a reduction of systemic cortisol, or a deregulation of its circadian production and not an increase in systemic cortisol (Figure 6). Such adrenal suppression could translate into an imbalance in MR/GR pathway activation in the RPE/choroid and subsequent deleterious consequences, leading to CSCR (Figure 6).

## 4. Material and Methods

### 4.1. Animals

All experiments were performed in accordance with the European Communities Council Directive 86/609/EEC and French national regulations and approved by local ethical committees (#4488 and #2541, Charles Darwin). Animals were kept in pathogen-free conditions with food, water and litter, and housed in a 12-hour/12-hour light/dark cycle.

Lewis rats (Ecully, France) between 8 and 12 weeks-old were divided into three groups, the control group that was only observed, the placebo group receiving saline sub-cutaneous injections, and the treatment group receiving subcutaneous dexamethasone injections. 

### 4.2. Experimental Scheme

A blood sample was obtained at baseline from all rats between 9 and 10 AM. The subcutaneous injections were administered daily at 9 AM in the morning for five consecutive days. Rats were then sacrificed at day 5 in the afternoon (between 4 and 5 PM, at a time expected to be the corticosterone peak [50]). Blood samples were obtained at the time of sacrifice, and the eyes were enucleated and dissected to recover intraocular fluids and ocular tissues [Figure 7]. Aqueous humor and vitreous from both eyes of each rat were pooled, whereas the iris and ciliary body complex, neural retina, choroid and RPE complex were carefully dissected and isolated separately from each eye. Samples were snap-frozen in liquid nitrogen and stored at −80 °C for adequate tissue preservation.

Dexamethasone phosphate sodium 4 mg/mL (Mylan, Paris) was injected subcutaneously with a dosage of 1mg/kg/day for five days in the treatment group. We chose a high dose of dexamethasone for 5 consecutive days because it was previously published that corticosterone drop secondary to adrenal atrophy was observed in rats already 5 days after systemic treatment [34]. This treatment regimen could thus mimics the adrenal insufficiency potentially occurring in patients treated with systemic, nasal, intraarticular administration, although high variability may occur in the HPA axis response in humans [62]. A saline solution (NaCl 0.9%) volume was added to normalize the injected volume to 200 μL. In the placebo group, rats received an injection of 200 μL of saline solution. The route and mode of administrations were identical. Rats in the treatment and control groups were anesthetized by intramuscular injection of ketamine (40 mg/kg) and Xylazine (4 mg/kg) at baseline and a blood sample was retrieved from the major vein at the base of the tail to obtain serum samples (Figure 7). Rats in the control group were sacrificed at baseline, blood samples were collected immediately after the death of the animal.

### 4.3. Corticosterone and Dexamethasone Quantification

The Elisa corticosterone kit (KGE009, R&D Systems, Abingdon Science Park Abingdon, OX14 3NB) 96-wells was used to determine the concentration of the total corticosterone, both bound and free, according to the manufacturer instructions. Rasd1 (Ras related dexamethasone induced 1) Elisa (OKEH06148, Aviva Systems Biology) was used to quantify indirectly dexamethasone in rat serum and ocular fluids as Rasd1 is a small GTPase protein that is induced by dexamethasone [63].

### 4.4. Quantitative PCR

Total RNA was isolated from tissues using the RNeasy Mini Kit (Cat. Nb. 74106, QIAGEN) including DNase I treatment (Cat. Nb. 79254, QIAGEN) according to manufacturer’s instructions. First-strand complementary DNA was synthesized from the total mRNA using random primers (ThermoFisher Scientific) and SuperScript II reverse transcriptase (ThermoFisher Scientific). Transcript levels were analyzed by quantitative real-time PCR performed in QuantStudio™ 5 Real-Time PCR (Applied Biosystems, Foster City, CA, USA) with SYBR Green detection. Delta CT threshold calculation was used for relative quantification of results.

We have analyzed the expression of the gene *Nr3c2* (encoding MR), *Nr3c1* (encoding GR), *11β-hsd1*, *11β-hsd2* in the neural retina, the RPE/choroid complex and the iris/ ciliary body complex.

In the neural retina, we tested the expression of genes known to be regulated by MR and GR activation in the retina such as *Scnn1A*, encoding the Na+ channel ENac-, the K+ channel *Kir4.1*, as well as the water channel *Aqp4* [33,64]. 

In RPE/ choroid complex, we quantified the expression of genes shown to be up- regulated by aldosterone or MR pathway activation in human RPE cells derives from iPS such as *Plaur* encoding the urokinase plasminogen activator receptor (uPAR) and *Fosl1* encoding the FOS like 1 AP-1 transcription factor subunit protein, *Ptx3* encoding pentraxin 3 and *Omg* that encodes the oligodendrocyte myelin glycoprotein and that was down-regulated by aldosterone or Cortisol+ RU486 [65]. We also tested genes showed to be upregulated in the RPE/choroid complex after intravitreal injection of aldosterone such as *Ngal* encoding lipocalin 2 and *Mmp9* and the Shroom Family Member 2 *Shroom2,* that is implicated in amiloride-sensitive sodium channel activity [66]. *Hprt1*, *Ubc*, *18S* were used as reference genes (Table 1). 

### 4.5. Statistical Analysis 

Quantitative data were expressed as mean ± SE. Statistical analysis was made using the Graphpad Prism 5 program (Graphpad Software, San Diego, CA, USA). P < 0.05 was considered significant. Groups were compared using Kruskal-Wallis test followed by a Dunn’s multiple comparison test. P< 0.05 was considered significant.

## 5. Conclusions

In conclusion, although systemic exposures to exogenous GCs have been considered as risk factors for the development of CSCR, the mechanisms of this risk remain poorly understood. The fact that intraocular injection of glucocorticoids does not increase the risk of CSCR suggests that the consequences of the HPA axis regulations for ocular tissues may be involved. Our results are in line with this hypothesis, and they suggest that it could be rather a lack of activation of the glucocorticoid pathway and an imbalance in favor of activation of the mineralocorticoid pathway in RPE/choroid, observed in response to the HPA axis brake that could favor CSCR. Further studies are needed to confirm this hypothesis.

## Figures and Tables

**Figure 1 ijms-23-01278-f001:**
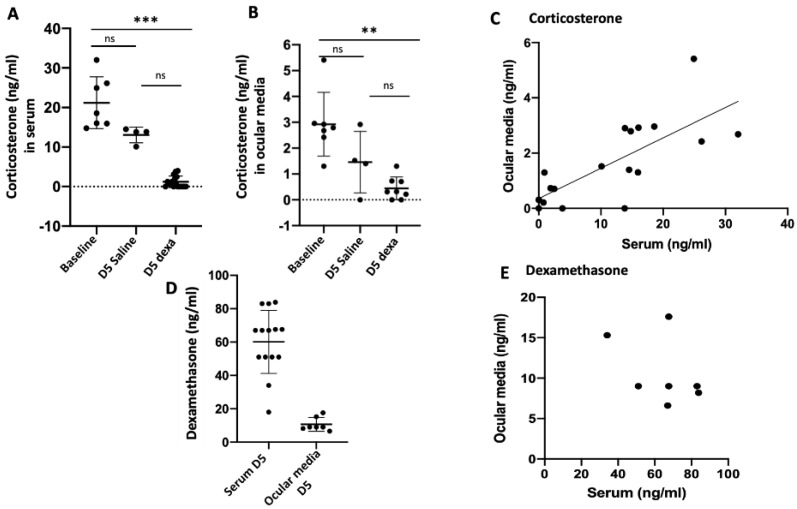
Corticosterone and dexamethasone levels in the rat serum and ocular media. Corticosterone in the serum (**A**) and ocular media (**B**) at baseline (*n* = 7) and after 5 days of daily injection of dexamethasone (D5) (*n* = 16) or saline (*n* = 5). Spearman correlation between corticosterone ocular and serum levels (r = 0.76, *p* = 0.0001) (**C**). Dexamethasone serum (*n* = 14) and ocular media (*n* = 7) after 5 days of daily injection of dexamethasone (**D**). Spearman correlation between dexamethasone ocular and serum levels (r = −0.28, *p* = 0.55) (**E**), ** *p* < 0.001, *** *p* < 0.0001, ns non significant.

**Figure 2 ijms-23-01278-f002:**
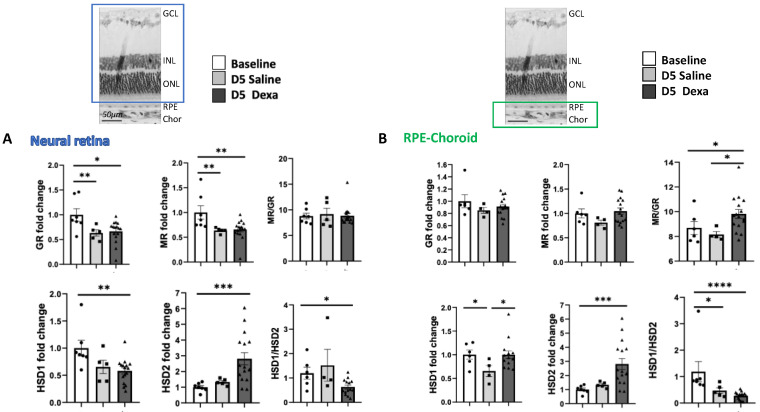
Corticoid receptors and hydroxysteroid dehydrogenase gene expression in the retina and RPE/choroid. (**A**) neural retina: *Nr3c1*, encoding GR, *Nr3c2* encoding MR and *11β-hsd1* and *2* expression at baseline (*n* = 7), after 5 days of daily injection of dexamethasone (*n* = 16), or saline (*n* = 6); (**B**) RPE/choroid: *Nr3c1*, encoding GR, *Nr3c2* encoding MR and *11β-hsd1*(HSD1*) and 2* (HSD2) expression at baseline (*n* = 7), after 5 days of daily injection of dexamethasone (*n* = 16), or saline (*n* = 6), * *p* < 0.05, ** *p* < 0.01, *** *p* < 0.001, **** *p* < 0.0001.

**Figure 3 ijms-23-01278-f003:**
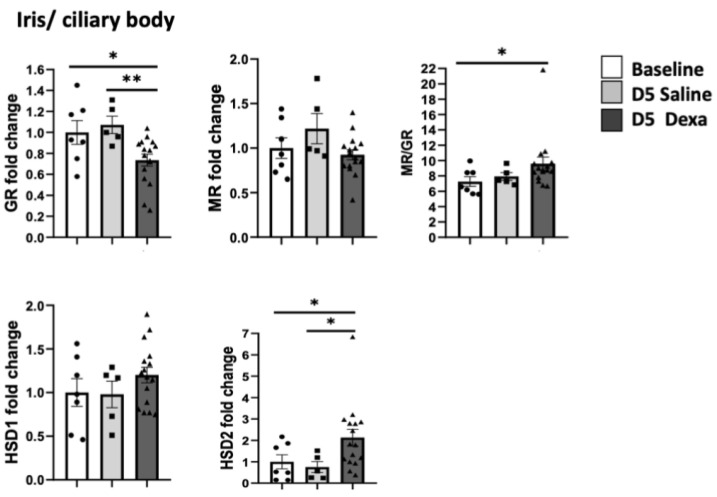
Corticoid receptors and hydroxysteroid dehydrogenase gene expression in the iris/ciliary body. *Nr3c1*, encoding GR, *Nr3c2* encoding MR and *11β-hsd1* and 2 expression at baseline (*n* = 7), after 5 days of daily injection of dexamethasone (*n* = 16), or saline (*n* = 6), * *p* < 0.05, ** *p* < 0.01.

**Figure 4 ijms-23-01278-f004:**
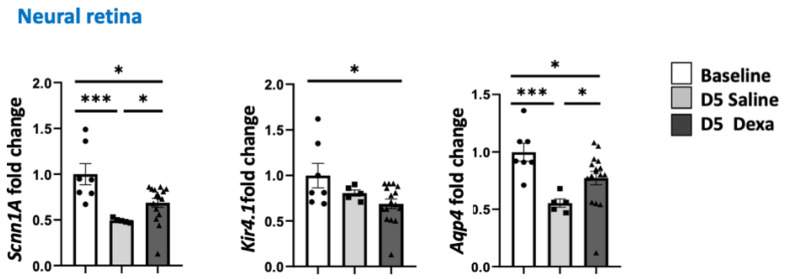
Regulation of MR-induced genes in the neural retina. *Scnn1A* encoding ENac-α Kir4.1 and Aqp4 expression at baseline (*n* = 7), after 5 days of daily injection of dexamethasone (*n* = 16), or saline (*n* = 6), * *p* < 0.05, *** *p* < 0.001.

**Figure 5 ijms-23-01278-f005:**
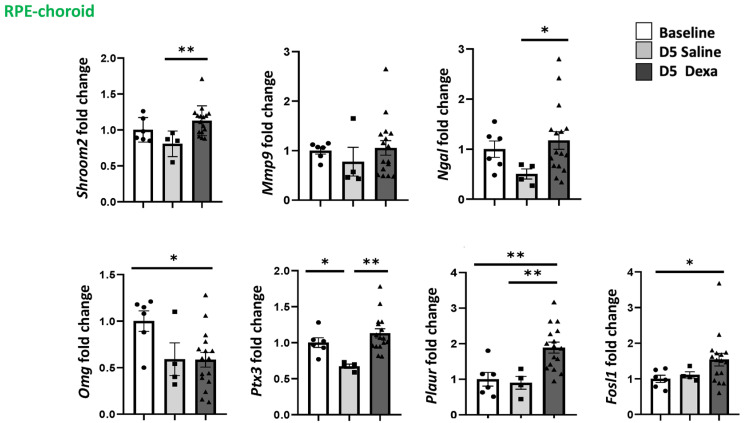
Regulation of MR-induced genes in the RPE/choroid. Expression of *Shroom 2*, *Ngal*, *Mmp9* and *Omg*, *Ptx3*, *Plaur* and *Fosl-1* in the RPE/choroid of rats at baseline (*n* = 7), after 5 days of daily injection of dexamethasone (*n* = 16) or saline (*n* = 6), * *p* < 0.05, ** *p* < 0.01.

**Figure 6 ijms-23-01278-f006:**
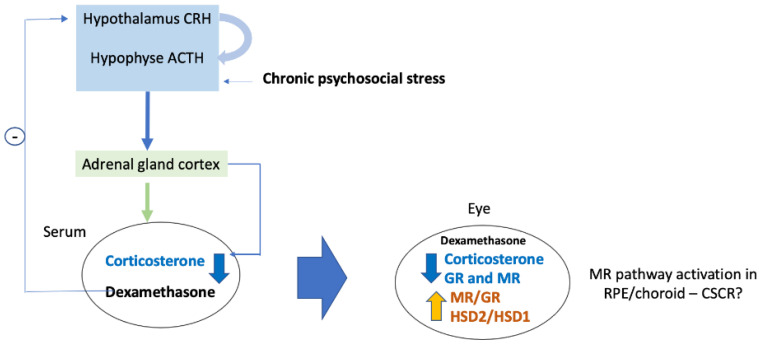
Schematic representation of the ocular consequences of HPA axis brake on ocular GR/MR pathways [34,49,61].

**Figure 7 ijms-23-01278-f007:**
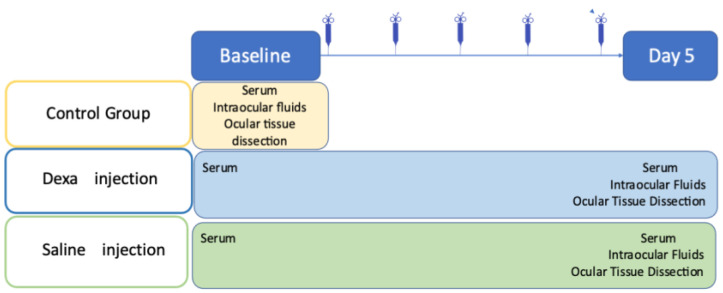
Schematic representation of the experiments. Serum was extracted from blood samples, intraocular fluids include aqueous humour and vitreous from each eye. Ocular tissue dissection separated the iris-ciliary body complex, the neural retina, the retinal pigment epithelium (RPE)-choroid complex or each eye separately.

**Table 1 ijms-23-01278-t001:** Shows the list of primers used in this study.

Gene Name	Forward/Reverse	Sequences
*18S*	Forward	TGCAATTATTCCCCATGAACG
*18S*	Reverse	GCTTATGACCCGCACTTACTGG
*Ubc*	Forward	ATCTAGAAAGAGCCCTTCTTGTGC
*Ubc*	Reverse	ACACCTCCCCATCAAACCC
*Hprt1*	Forward	GCGAAAGTGGAAAAGCCAAGT
*Hprt1*	Reverse	GCCACATCAACAGGACTCTTGTAG
*Nr3c1*	Forward	AACATGTTAGGTGGGCGTCAA
*Nr3c1*	Reverse	GGTGTAAGTTTCTCAAGCCTAGTATCG
*Nr3c2*	Forward	TAAGTTTCCCCACGTGGTTC
*Nr3c2*	Reverse	ATCCACGTCTCATGGCTTTC
*11b-Hsd1*	Forward	GAAGAAGCATGGAGGTCAAC
*11b-Hsd1*	Reverse	GCAATCAGAGGTTGGGTCAT
*11b-Hsd2*	Forward	TGCTGGCTGGATCGCGTTGTC
*11b-Hsd2*	Reverse	CACAGTGGCCAGCACCGTGAA
*Aqp4*	Forward	CGGTTCATGGAAACCTCACT
*Aqp4*	Reverse	CATGCTGGCTCCGGTATAAT
*Kir4.1*	Forward	CAAAGAAGAGGGCTGAGACG
*Kir4.1*	Reverse	TTGAGCCGAATATCCTCACC
*Scnn1A*	Forward	TCATGCTGCTACGCCGGTTCC
*Scnn1A*	Reverse	TCCATCAGTTTACAAGGGAG
*Ngal*	Forward	TCACCCTGTACGGAAGAACC
*Ngal*	Reverse	GGTGGGAACAGAGAAAACGA
*Mmp9*	Forward	CTGCCTGCACCACTAAAGG
*Mmp9*	Reverse	GAAGACGAAGGGGAAGACG
*Omg*	Forward	CAACACAATGCAGCTGAGCA
*Omg*	Reverse	CCATCAAAGCCATAATGTCGTCT
*Ptx3*	Forward	TACCCGCAGGCTGTGAAAC
*Ptx3*	Reverse	GGGTTCCACTTGGTGCCATA
*Plaur*	Forward	CTGAAGTGCTGCAACTTCAC
*Plaur*	Reverse	AGCACATCTAAGCCTGTAGC
*Fosl1*	Forward	GAGAAAATATGTCCCTCTGC
*Fosl1*	Reverse	TTCTAGGCTAGTTAAAGGGC
*Shroom2*	Forward	CCTATTATAGCACATCAGCC
*Shroom2*	Reverse	TGAGCTCTTGCTTCTTAATG

## Data Availability

All raw data are available upon reasonable request.

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
