# Peer review of "Chronic Systemic Dexamethasone Regulates the Mineralocorticoid/Glucocorticoid Pathways Balance in Rat Ocular Tissues"

_ijms, 2022, doi:10.3390/ijms23031278_

Round 1

Reviewer 1 Report

Behar-Cohen et al. set out to study the corticosterone ocular levels and the corticoid receptors balance upon brake of the hypothalamic-pituitary-adrenal axis after systemic dexamethasone treatment in rats. They found indication that systemic GCs and endogenous corticosterone influence the expression of the gluco and mineralocorticoid receptors in different ocular tissues and promote an imbalance towards a hyperactivation of the mineralocorticoid pathway which is in line with previous work. They also try to draw a link with CSCR especially because of the changes of GC and MC signaling in the choroidal tissue they observed.

As such, the study addresses important issues of interest for a broader readership as dexamethasone is widely used and unwarranted side effects including that in the eye need to be better understood. However, I am a bit puzzeled about how data are presented and discussed. My main concern is how the authors present and discuss the finding that saline injections at five consecutive days has a very similar, depending on the readout at times even stronger effect than the actual dexamethasone injection. I have no issues with the data as such and it is clear that such a treatment paradigm can cause stress and associated regulatory loops in the animals and therewith changes in the activity of the HPA axis, which kind of could mimic the administration of exogenous GC, but this should be way clearer presented and discussed. Please revise the manuscript extensively and clearly delineate limits of the experimental design. Wouldn’t one at least expect a synergistic effect of Dexamethasone treatment and the stress induced by the treatment paradigm per se? This is clearly not the case –why?

Other more specific points:

  • What is the rationale of choosing a 5 days treatment regimen? A brief justification would be nice e.g. in relation to the occurrence of CSCR as unwarranted side effect.
  • Figure 6 seems to lack the data for Omg, Ptx3, Plaur and Fosl-1. Please add.
  • Figure 7 – this scheme does not only base on data generated in the present study, accordingly proper literature should be cited that support this summary.
  • Why is the effect of saline on ENAC and AQP4 expression more pronounced than that of Dexa? Seems to be an effect independent from GC/MC pathways then, correct?
  • Generally, a validation of gene expression changes at protein level (e.g. immunostainings) would largely improve the scope of the paper as proteins and their correct localization in the cell define the actual function.
  • AQP4 and Kir4.1 are very specific glial genes – this not mentioned at all, but should be discussed since their downregulation could impact the retinal volume homeostasis and thus could cause retinal edema. Was something like this observed in the experimental animals or in CSCR patients?
  • The authors check for several target genes of the GC/MC pathways (Shroom 2, Ngal, Mmp9 and Omg, Ptx3, Plaur and Fosl-1), but do not (Shroom, Ngal, Mmp9, Ptx3) or only very briefly (Omg, Plaur, Fosl-1) discuss a functional implication of their respective dysregulation. A brief discussion thereof is recommended to justify this set of experiments.

Minor points:

  • The text contains some typos, wrong symbols (e.g. for ENac-alpha) and lots of extra spaces between words that should be deleted.
  • The legend for Diagram 1 says “Schematic representation of the experience.” Replace “experience” with “experiments!
  • The figure legends are in a strange format and hard to distinct from the main text. Please revise.

Author Response

Behar-Cohen et al. set out to study the corticosterone ocular levels and the corticoid receptors balance upon brake of the hypothalamic-pituitary-adrenal axis after systemic dexamethasone treatment in rats. They found indication that systemic GCs and endogenous corticosterone influence the expression of the gluco and mineralocorticoid receptors in different ocular tissues and promote an imbalance towards a hyperactivation of the mineralocorticoid pathway which is in line with previous work. They also try to draw a link with CSCR especially because of the changes of GC and MC signaling in the choroidal tissue they observed.

As such, the study addresses important issues of interest for a broader readership as dexamethasone is widely used and unwarranted side effects including that in the eye need to be better understood.

Thank you for the encouraging comments. We agree that the topic is not simple and requires extensive studies. This one is only a first step.

However, I am a bit puzzeled about how data are presented and discussed. My main concern is how the authors present and discuss the finding that saline injections at five consecutive days has a very similar, depending on the readout at times even stronger effect than the actual dexamethasone injection. I have no issues with the data as such and it is clear that such a treatment paradigm can cause stress and associated regulatory loops in the animals and therewith changes in the activity of the HPA axis, which kind of could mimic the administration of exogenous GC, but this should be way clearer presented and discussed. Please revise the manuscript extensively and clearly delineate limits of the experimental design.

We have modified the introduction, presentation and discussion regarding the low corticosterone levels in rats that received saline injection to make it clearer.

In the introduction, we have added:

But contrarily to the other GCs-induced side effects, the route of GCs administration influences the risk of CSCR.  The nasal, articular, oral and dermal routes favor CSCR [12–21] but not the high and sustained intraocular GCs [22,23]. This clinical observation led us to hypothesize that it is not the increase in ocular  GCs levels but rather a consequence of systemic corticosteroids on ocular corticosteroids that may be the link between CSCR  and corticosteroid therapy”.

In the discussion

Corticosterone levels also decreased, although to a lesser extent after repeated daily injection of saline in awaken rats, submitted to repeated restraint stress at fixed hour for 5 days, which corresponds to a chronic but familiar stressor. Indeed, under such stress conditions, corticosterone was shown to decrease, as compared to unfamiliar stressor that could have a reverse effect on corticosterone levels  [45,46]. Such low corticosterone levels  also suggested by low salivary amylase activity in rats submitted to similar stressor [47] indicate changes in the activity of the HPA axis in response of a chronic stress, which could mimic the administration of exogenous GC.”

And

In the ophthalmic literature, stressful psychological events, that favor the occurrence of CSCR episodes[11], have been intuitively linked to high systemic cortisol, making the link with GCs-induced CSCR. Interestingly, no study could identify high serum cortisol levels in CSCR patients and hair-cortisol was not increased in this population[66]. We postulate that under chronic stress conditions, rumination and depression state, that is frequently observed in CSCR patients[67], and that have been associated with a reduced activity of the HPA axis [53,68,69], low endogenous cortisol could favor the occurrence of the disease in predisposed patients. In other words, the link between exogenous systemic GCs treatment and psychological stress in patients with CRSC could be a reduction of systemic cortisol, or a deregulation of its circadian production and not an increase of cortisol [Figure 7]. Such adrenal suppression could translate into imbalance in MR/GR pathway activation in the RPE/ choroid and subsequent deleterious consequences, leading to CSCR.”

Wouldn’t one at least expect a synergistic effect of Dexamethasone treatment and the stress induced by the treatment paradigm per se? This is clearly not the case –why?

The fact that saline induces changes that in some of the readouts could be even stronger than the dexamethasone could be explained by the fact that endogenous corticosterone and dexamethasone do not induce exactly the same regulations. In both cases, serum and ocular corticosterone levels are low, but in case of dexamethasone treatment, dexamethasone enters into the eye and may have some counteracting effects. It is thus not obvious that they should have synergistic effects on all the regulations.

Other more specific points:

  • What is the rationale of choosing a 5 days treatment regimen? A brief justification would be nice e.g. in relation to the occurrence of CSCR as unwarranted side effect.

Our hypothesis was that CSCR could result not from exogenous excess but to the consequence of GCs on the endogenous production of corticoids by the adrenal gland, in other word on the adrenal insufficiency induced by high dose of systemic glucocorticoids.

We have used 5 days treatment based on the old knowledge that atrophy of the adrenal gland is observed 5 days after systemic administration of GCs at high dose, which is the basis of the statement that any patient who received high dose of GCs for a week is at risk of adrenal insufficiency.

We also used old papers showing adrenal atrophy after 4-8 days after dexamethasone treatment [Leśniewska, B.; Nowak, K. W.;Malendowicz, L. K. (1992). Dexamethasone-Induced Adrenal Cortex

Atrophy and Recovery of the Gland from Partial, Steroid-Induced Atrophy. Experimental and Clinical Endocrinology & Diabetes, 100(6), 133–139. doi:10.1055/s-0029-1211193]

Longer treatment could have suppressed corticosterone more radically but it would have been potentially with many other side effects that could have interfered with our analysis.

The exact link and timing between CSCR and GCs intake is poorly defined, due to the fact that CSCR is symptomatic only when the retinal serous detachment includes the macula. Patients are often diagnosed with CSCR and with evidence of previous and/ or multiple episodes of detachments.

We added the following sentence in the material and method:

“We chose a high dose of dexamethasone for 5 consecutive days because it was previously published that corticosterone drop secondary to adrenal atrophy was observed in rats already 5 days after systemic treatment [36].  This treatment regimen could thus mimics the adrenal insufficiency potentially occurring in patients treated with systemic, nasal, intraarticular administration, although high variability may occur in the HPA axis response in humans [37].” Ref 36 and 37 were added

  • Figure 6 seems to lack the data for Omg, Ptx3, Plaur and Fosl-1. Please add.

We are sorry for the mistake. The data were added.

  • Figure 7 – this scheme does not only base on data generated in the present study, accordingly proper literature should be cited that support this summary

This is right. We have added several review papers of the suppression of the HPA axis by corticoids and psychosocial stress.

Adam EK, Quinn ME, Tavernier R, McQuillan MT, Dahlke KA, Gilbert KE.Diurnal cortisol slopes and mental and physical health outcomes: A systematic review and meta-analysis. Psychoneuroendocrinology. 2017 Sep;83:25-41. doi: 10.1016/j.psyneuen.2017.05.018. Epub 2017 May 24.

Chida Y, Hamer M. Chronic psychosocial factors and acute physiological responses to laboratory-induced stress in healthy populations: a quantitative review of 30 years of investigations. Psychol Bull. 2008 Nov;134(6):829-85. doi: 10.1037/a0013342.

B LeÅ›niewska  1 , K W Nowak, L K Malendowiczv Dexamethasone-induced adrenal cortex atrophy and recovery of the gland from partial, steroid-induced atrophy Exp Clin Endocrinol 1992;100(3):133-9. doi: 10.1055/s-0029-1211193.

Please, advise if other ref should be cited.

  • Why is the effect of saline on ENAC and AQP4 expression more pronounced than that of Dexa? Seems to be an effect independent from GC/MC pathways then, correct?

In the neural retina, both GR and MR are down-regulated without imbalance in the MR/GR ratio. In addition, low corticosterone levels are measured in the ocular media. In case of saline, there is also no dexamethasone to compensate for the low GR pathway activation. ENac-alpha and AQP4 are highly sensitive to MR and GR activation, that both induces upregulation of the genes,  as shown in a previous paper (Zhao et al Faseb 2010).  It is thus expected that ENac-alpha, AQP4 are down-regulated and even lower in saline treated rats. The low levels of dexamethasone in the eye might have partially compensate for the lack of corticosterone.

Yes, we agree that the effect on ENac-alpha and AQP4 could also result from other mechanisms than the GR/MR balance, i.e. the salt contained in the saline might have direct effects on retina ion and sodium channel expression. We are now conducting additional extensive experiments to evaluate the effects of various doses of vehicles with variable concentrations of salt, since it was not known that systemic salt intake could modify the expression of ion channels in the retina, since this has not been explored yet.

This has been added in the discussion as follows:

“In the neural retina, the Nr3c2/ Nr3c1 balance stayed stable after dexamethasone treatment, although both GR and MR were down-regulated which was  reflected by the down-regulation of genes such as Scnn1A (encoding ENac-alpha), Kir4.1 and Aqp4that are up-regulated by both GR and MR activation in glial Müller cells and in rat retinal explants [39]. In dexamethasone treated rats, the low levels of ocular dexamethasone could have partly counterbalanced the downregulation of Scnn1A, Kir4.1and Aqp4. It cannot be excluded that other mechanisms unrelated to GR/MR pathway and directly linked to systemic sodium levels could have played a role in the ion and water channels regulations observed in the neural retina of saline injected rats. This hypothesis requires further explorations.”

  • Generally, a validation of gene expression changes at protein level (e.g. immunostainings) would largely improve the scope of the paper as proteins and their correct localization in the cell define the actual function.

We agree that protein levels and / or immunolocalization would improve our understanding.

Another more extensive experiment has been planned and to analyze various concentrations of salt on the expression of ion and water channels in the retina and the effects of other treatments. Results will not be available before several months and are not exactly in the scope of the present paper.

Here, we found no change in the GR/MR balance in the neural retina and particularly no unbalance towards MR overactivation despite a significant reduction of MR and GR expression. We used ENac, Kir4.1 and AQP4 to confirm the low activation of GR and MR without imbalance.

Indeed, our previous work showed that in the neural retina and in glial Müller cells, both MR and GR activation by aldosterone or dexamethasone induced an over expression of  ENac, Kir4.1 and AQP4, that was associated with neural retinal increased thickness due to Müller cells swelling.

In our experiments, there is a reduction in the expression of ENac, Kir4.1 and AQP4, in both saline and dexamethasone treated rat neural retina, which is in line with low GR and MR activation.

In eyes with dexamethasone, the low ocular levels could explain that expression of ion and water channels are higher, although not increased as compared to controls.  We thus are not expecting any retinal edema resulting from such expression changes.

In case of chronic treatment with dexamethasone and eventually with other GCs in humans, we observe changes in the RPE/ choroid complex such as CSCR but with minimal or no change in the neural retina. This has been stressed along the discussion.

We also have added in the discussion the following paragraph:

We acknowledge weaknesses in our study. Aldosterone could not be measured due to the low volume of ocular media and methods used in this study. Further studies will use mass spectrometry to give a broader analysis of all neurosteroids and aldosterone in the ocular media. We have not yet validated the regulated target genes at the protein level using either immunohistochemistry or western-blotting, which will be the subject of another study, that will include many other controls including various concentrations of salt in the vehicle”.

  • AQP4 and Kir4.1 are very specific glial genes – this not mentioned at all, but should be discussed since their downregulation could impact the retinal volume homeostasis and thus could cause retinal edema. Was something like this observed in the experimental animals or in CSCR patients?

Please see above

  • The authors check for several target genes of the GC/MC pathways (Shroom 2, Ngal, Mmp9 and Omg, Ptx3, Plaur and Fosl-1), but do not (Shroom, Ngal, Mmp9, Ptx3) or only very briefly (Omg, Plaur, Fosl-1) discuss a functional implication of their respective dysregulation. A brief discussion thereof is recommended to justify this set of experiments.

We are sorry, all the results were added.

Extensive discussion on the significance of such gene deregulation in the RPE and their relation with CSCR is available in Canonica, J.; Zhao, M.; Favez, T.; Gelizé, E.; Jonet, L.; Kowalczuk, L.; Guegan, J.; Le Menuet, D.; Viengchareun, S.; Lombès, M.; et al. Pathogenic Effects of Mineralocorticoid Pathway Activation in Retinal Pigment Epithelium. Int J Mol Sci 2021, 22, 9618, doi:10.3390/ijms22179618.

We have added in the discussion, some comments on the significance of these gene regulation as follows:

“To further confirm this imbalance towards MR activation, we tested the expression of genes that we previously identified as MR-target genes in human RPE cells derived from iPSc using an unbiased transcriptomic approach [41]. In the RPE/choroid, Plaur and Fosl1 were upregulated and Omg was down-regulated as previously shown when RPE cells, stimulated by aldosterone or by cortisol + GR antagonists to mimic the effects of cortisol on MR [41].  FOS like 1 AP-1 transcription factor subunit protein controls the expression of Plaur (uPAR) [54]. The uPAR is expressed at the basolateral membrane of RPE cells [55] and activation of uPA-uPAR pathways favors choroidal neovascularization [56] and induces RPE epithelial to mesenchymal transition and proliferation [57]. The upregulation of this pathway could be involved in the RPE barrier breakdown, that is one of the cardinal features of CSCR. Shroom2 is a protein involved in the pigmentation and of RPE cells [58] and pigmentary changes are part of CSCR phenotype. The down regulation of OMG in RPE cells, which encodes oligodendrocyte-myelin glycoprotein, could indicate a role for the RPE in the maintenance of choroidal nerves, which control the choroidal blood flow [49] through a secreted isoform of OMG [50]. Alteration of choroidal vasculature is the other cardinal feature for CSCR.

Other genes encoding proteins involved in inflammation and oxidative stress such as pentraxin 3 and lipocalin 2 were mostly down regulated in RPE/ choroid from saline injected rats. Interestingly, we found that systemic low lipocalin levels could be a biomarker for CSCR in cohorts of patients that were not under GCs treatments[59]. Since dexamethasone is a known inducer of pentraxin 3 and lipocalin 2 expression[60], a partial counter-balance could have compensated their reduced expression due to low GR/MR expression in eyes from dexamethasone treated rats.

These results indicate that chronic systemic dexamethasone treatment causes an overactivation of the MR pathway, specifically in the RPE/ choroid complex and not in the retina. In a mouse transgenic model that overexpresses the human MR, we observed pathologic changes in the RPE/ choroid that mimic the pachychoroid phenotype, frequently associated in humans with CSCR, without significant changes in the neural retina [34,61,62].”

Minor points:

  • The text contains some typos, wrong symbols (e.g. for ENac-alpha) and lots of extra spaces between words that should be deleted.
  • The legend for Diagram 1 says “Schematic representation of the experience.” Replace “experience” with “experiments!
  • The figure legends are in a strange format and hard to distinct from the main text. Please revise.

Thank you, we have corrected the minor points

Reviewer 2 Report

The goal of the manuscript submitted by Zola et al. is to test the hypothesis that the hypothalamic-pituitary-adrenal axis (HPA) brake could be involved in CSCR pathogenesis, and to evaluate the mineralocorticoid/ glucocorticoid pathways balance in ocular tissues after dexamethasone treatment. To accomplish that, rats received systemic injection of either dexamethasone or saline daily for 5 days. Baseline levels of corticosterone were measured by Elisa at baseline and at 5 days in the serum and the ocular media and dexamethasone levels were measured at 5 days in the serum and ocular media. The expression level of Nr3c1, Nr3c2, hsd1 and 2 were measured in the neural retina, retinal pigment epithelium (RPE)/ choroid and iris/ ciliary body.

Clinical Data showed that Patients affected by acute CSC treated with MR   antagonists achieved greater and faster resolution of functional and anatomical manifestation of CSCR. The authors should consider including the manuscripts from Zucchiatti et. al. (Ophthalmol Ther (2018) 7:109–118) and Borrelli at al. (J. Clin. Med. 2019, 8, 1271) in the discussion.

Author Response

Thank you for the positive evaluation of our Ms, we have added the suggested references

Round 2

Reviewer 1 Report

The authors addressed most issues of my review.

Saw the figure with all data now included. Fine for me. The figure would of course be nicer if all x-axis in a lane would be aligned - the figure looks still a bit messy, but this pure cosmetics.

Author Response

We have revised the figures and completely re written the abstract

We hope that now the paper will be acceptable for publication 
